# Confidence Interval Estimation for Precipitation Quantiles Based on Principle of Maximum Entropy

**DOI:** 10.3390/e21030315

**Published:** 2019-03-22

**Authors:** Ting Wei, Songbai Song

**Affiliations:** College of Water Resources and Architectural Engineering, Northwest A&F University, Yangling 712100, China

**Keywords:** principle of maximum entropy, quantile estimation, confidence interval, Monte Carlo simulation, precipitation frequency analysis

## Abstract

The principle of maximum entropy (POME) has been used for a variety of applications in hydrology, however it has not been used in confidence interval estimation. Therefore, the POME was employed for confidence interval estimation for precipitation quantiles in this study. The gamma, Pearson type 3 (P3), and extreme value type 1 (EV1) distributions were used to fit the observation series. The asymptotic variances and confidence intervals of gamma, P3, and EV1 quantiles were then calculated based on POME. Monte Carlo simulation experiments were performed to evaluate the performance of the POME method and to compare with widely used methods of moments (MOM) and the maximum likelihood (ML) method. Finally, the confidence intervals *T*-year design precipitations were calculated using the POME for the three distributions and compared with those of MOM and ML. Results show that the POME is superior to MOM and ML in reducing the uncertainty of quantile estimators.

## 1. Introduction

One of the objectives of hydrological frequency analysis is to estimate the magnitude of a hydrologic event with a given return period [1]. Due to limited data records, inappropriate assumption regarding the parent distribution, and errors associated with parameters estimation, there are inevitably uncertainties in this estimation [2,3]. Hence, a point estimate of quantile corresponding to a desired return period is usually not enough because it cannot adequately describe the reliability of the estimation. Confidence interval is a convenient approach to quantifying the uncertainty of the estimates and provides more information than just a point estimate or its standard error [4]. 

The calculation of confidence interval requires a standard error of quantile estimator, and several methods have been proposed for determining such standard error. Hoshi and Barges derived the expressions for calculating the sampling variances and covariances of log-Pearson type 3 (P3) distribution parameters as well as the sampling variance of *T*-year flood event using the method of moments (MOM) [5]. Condie gave the maximum likelihood estimators for the parameters of a log-Pearson type 3 distribution, derived the expressions for asymptotic standard error of a *T*-year event, and concluded that the maximum likelihood (ML) method is markedly superior to MOM in the estimation of asymptotic standard error of *T*-year event [6]. Lu and Stedinger derived the simple formulas for estimating the asymptotic variance of probability weighted moments (PWM) quantile estimators for generalized extreme value (GEV) distribution when the location and scale parameters were estimated with a fixed regional shape parameter or all three parameters were estimated [4]. Phien derived the explicit formulas for the variances and covariances of the parameter estimates of log-Pearson type 3 distribution when the method of direct and mixed moments was used for parameter estimation [7]. The confidence intervals of MOM and ML quantile estimators for log-Gumbel, Weibull, and generalized logistic distribution distributions have also been investigated [8,9,10]. 

Shannon defined the concept of entropy as a measure of uncertainty of a random variable or its probability distribution [11]. Jaynes later formulated the principle of maximum entropy (POME), which provides a rational approach to choosing the most unbiased probability distribution for hydrologic frequency analysis [12]. Sonuga developed a minimally biased probability distribution appropriate for hydrologic frequency analysis in the absence of a large amount of data [13]. Singh developed a procedure for derivation of a number of frequency distributions used in hydrology using POME [14]. Lu derived the generalized distribution for flood and extreme rainfall frequency analysis, and she concluded that the entropy-based generalized distributions are superior or comparable to other traditional distributions [15,16]. POME also provides a way to estimate parameters of a given distribution from the specified constraints. Singh summarized the entropy method for parameter estimation for the commonly used distributions and indicated that the entropy method is reasonable and efficient for parameter estimation [17]. The POME-based parameter estimations for some other distributions have also been derived [18,19,20]. In recent years, an integration of entropy and copula has been developed to construct joint distribution function capable of bivariate flood and drought analysis as well as streamflow simulation [21,22,23]. 

For the estimation of the POME-based variance, Phien provided the formulas for calculating the approximate variances of the parameter estimators and *T*-year event for the extreme value type-1 (EV1) distribution and P3 distributions [24,25]. Through applications of the formulas to simulated data, he concluded that the approximate variance of estimates of the *T*-year event are of sufficient accuracy. However, there are no follow-up studies on the POME-based confidence interval estimation of quantile estimators.

The objective of this study is therefore to apply POME further in the estimation of confidence intervals of quantile estimators. The Monte Carlo simulation was carried out to evaluate the performance of POME in the calculation of confidence intervals based on simulated data sets. Then, the hamma, P3, and EV1 distributions were used to fit the observed annual precipitation series. The distribution parameters and confidence intervals of annual precipitation quantiles for different return periods were estimated using POME, MOM, and ML. Finally, the confidence intervals based on different methods were compared. 

## 2. Confidence Interval Estimation of the Quantile

### 2.1. Estimation of Quantile

A general form for calculating ${\hat{x}}_{T}$ of a given distribution can be written in terms of the distribution moments and the frequency factor $K_{T}$ [26]:(1)$${\hat{x}}_{T}={\hat{\mu }}_{1}^{\prime }+K_{T}\sqrt{{\hat{\mu }}_{2}}$$
where ${\hat{\mu }}_{1}^{\prime }$ and ${\hat{\mu }}_{2}$ are the mean and the standard error of the population, respectively, and they equal the sample moments only when the MOM is used for parameter estimation; $K_{T}$ is the frequency factor specific to the chosen distribution, which can be derived from the distribution parameters, sample size, and return period *T* or cumulative probability of exceedance of the design event. Expressions of $K_{T}$ for different distributions are commonly given in statistics texts [1].

### 2.2. Calculation of Confidence Interval

The standard error and confidence interval are two measures to describe the precision of a statistical quantity, such as the *T*-year quantile estimator ${\hat{x}}_{T}$. The (1−α) confidence interval for ${\hat{x}}_{T}$ is approximated by [27]:(2)$${\hat{x}}_{L}={\hat{x}}_{T}\pm u_{1-\frac{\alpha }{2}}s_{T}$$
where ${\hat{x}}_{L}$ is the confidence interval; $u_{1-\frac{\alpha }{2}}$ is the quantile of the standard normal distribution for confidence levels equal to $1-\frac{\alpha }{2}$; ${\hat{x}}_{T}$ is the design value for the return period *T*; $s_{T}$ is the standard error of ${\hat{x}}_{T}$, which can be expressed as [27]:(3)$$\begin{array}{l} s_{T}^{2}=\mathrm{var}\left({\hat{x}}_{T}\right)={\left(\frac{\partial x_{T}}{\partial \theta_{1}}\right)}^{2}\mathrm{var}\left({\hat{\theta }}_{1}\right)+{\left(\frac{\partial x_{T}}{\partial \theta_{2}}\right)}^{2}\mathrm{var}\left({\hat{\theta }}_{2}\right)+{\left(\frac{\partial x_{T}}{\partial \theta_{3}}\right)}^{2}\mathrm{var}\left({\hat{\theta }}_{3}\right)\\ +2\frac{\partial x_{T}}{\partial \theta_{1}}\frac{\partial x_{T}}{\partial \theta_{2}}\mathrm{cov}\left({\hat{\theta }}_{1},{\hat{\theta }}_{2}\right)+2\frac{\partial x_{T}}{\partial \theta_{2}}\frac{\partial x_{T}}{\partial \theta_{3}}\mathrm{cov}\left({\hat{\theta }}_{2},{\hat{\theta }}_{3}\right)+2\frac{\partial x_{T}}{\partial \theta_{1}}\frac{\partial x_{T}}{\partial \theta_{3}}\mathrm{cov}\left({\hat{\theta }}_{1},{\hat{\theta }}_{3}\right) \end{array}$$
where ${\hat{\theta }}_{i},i=1,2,3$ denotes the estimators of either moments or distribution parameters; $\mathrm{var}\left({\hat{\theta }}_{i}\right)$ is the variance of $\theta_{i}$; $\mathrm{cov}\left({\hat{\theta }}_{i},{\hat{\theta }}_{j}\right)$ is the covariance of ${\hat{\theta }}_{i}$ and ${\hat{\theta }}_{j}$; i,j=1  ,2,  3.

In this paper, the MOM, ML, and POME were considered, and the asymptotic variances estimated by these methods are described below.

#### 2.2.1. Method of Moments (MOM)

The MOM asymptotic variance of ${\hat{x}}_{T}$ for a three-parameter distribution is given by [27]:(4)$$\begin{array}{l} s_{T}^{2}=\frac{\mu_{2}}{n}\{1+\gamma_{1}K_{T}+\frac{K_{T}^{2}}{4}\left(\gamma_{2}-1\right)+\frac{\partial K_{T}}{\partial \gamma_{1}}\left[2\gamma_{2}-3\gamma_{1}^{2}-6+K_{T}\left(\gamma_{3}-\frac{3}{2}\gamma_{1}\gamma_{2}-\frac{5}{2}\gamma_{1}\right)\right]\\ \;\;\;\;\;\;\;\;\;\;\;\;+{\left(\frac{\partial K_{T}}{\partial \gamma_{1}}\right)}^{2}\left(\gamma_{4}-3\gamma_{1}\gamma_{3}-6\gamma_{2}+\frac{9}{4}\gamma_{1}^{2}\gamma_{2}+\frac{35}{4}\gamma_{1}^{2}+9\right)\} \end{array}$$
where $\gamma_{j},j=1,2,3,4$ are the cumulants.

For a two-parameter distribution, the frequency factor $K_{T}$ does not depend on $\gamma_{1}$, then $\partial K_{T}/\partial \gamma_{1}=0$ in the above equation and the expression simplifies to: (5)$$s_{T}^{2}=\frac{\mu_{2}}{n}\left[1+\gamma_{1}K_{T}+\frac{K_{T}^{2}}{4}\left(\gamma_{2}-1\right)\right]$$

#### 2.2.2. Maximum Likelihood (ML) Method 

ML is a probability distribution-related method that requires the log-likelihood function of the probability density function (pdf) of a specific distribution. The ML parameters estimators of the commonly used distributions in hydrology are available in the literature [1].

The asymptotic variance and covariance terms for the ML parameter estimators are the elements of the inverse of the information matrix **I** [28]:(6)$$\begin{array}{l} \left[\begin{array}{ccc} \mathrm{var}\left({\hat{\theta }}_{1}\right) & \mathrm{cov}\left({\hat{\theta }}_{1},{\hat{\theta }}_{2}\right) & \mathrm{cov}\left({\hat{\theta }}_{1},{\hat{\theta }}_{3}\right)\\ & \mathrm{var}\left({\hat{\theta }}_{2}\right) & \mathrm{cov}\left({\hat{\theta }}_{2},{\hat{\theta }}_{3}\right)\\ & & \mathrm{var}\left({\hat{\theta }}_{3}\right) \end{array}\right]\\ ={\left[\begin{array}{ccc} E\left(-\frac{\partial^{2}\log L}{\partial \theta_{1}^{2}}\right) & E\left(-\frac{\partial^{2}\log L}{\partial \theta_{1}\partial \theta_{2}}\right) & E\left(-\frac{\partial^{2}\log L}{\partial \theta_{1}\partial \theta_{3}}\right)\\ & E\left(-\frac{\partial^{2}\log L}{\partial \theta_{2}^{2}}\right) & \;\;E\left(-\frac{\partial^{2}\log L}{\partial \theta_{2}\partial \theta_{3}}\right)\\ & & \;\;E\left(-\frac{\partial^{2}\log L}{\partial \theta_{3}^{2}}\right) \end{array}\right]}^{-1}=I^{-1} \end{array}$$

Differentiating Equation (1) with parameters $\theta_{1}$, $\theta_{2}$, and $\theta_{3}$, one obtains the derivatives of $x_{T}$ with respect to $\theta_{1}$, $\theta_{2}$, and $\theta_{3}$. Substituting the derivative terms and the asymptotic variances and covariances in Equation (6) into Equation (3) yields the asymptotic variance of the ML quantile estimators. Finally, the confidence interval of quantile estimators can be calculated by using Equation (2).

#### 2.2.3. Principle of Maximum Entropy (POME) Method

POME involves essentially five steps in the estimation of the distribution parameters: (1) specification of constraints from the given information; (2) derivation of the probability density function of the maximum entropy distribution; (3) derivation of the relationship between Lagrange multipliers and constraints; (4) derivation of the relationship between Lagrange multipliers and distribution parameters; and (5) derivation of the relationship between distribution parameters and constraints [17,19].

The constraints in POME can be expressed in terms of moments, therefore, the variance and covariances of the parameters can be obtained from the relationship between the variance and covariances of the moments and that of the parameter estimates. Let *P*, *Q,* and *R* denote the three moments, thus one can approximately write the vector of variance and covariances of *P*, *Q,* and *R* of a three-parameter distribution as [24]:(7)$$V_{M}=\;DV_{P}$$
where $V_{M}$ and $V_{P}$ are the vectors of variance and covariances of the moments and parameter estimators, respectively:(8)$$V_{M}=\left[\begin{array}{c} \mathrm{var}\left(P\right)\\ \mathrm{var}\left(Q\right)\\ \mathrm{var}\left(R\right)\\ \mathrm{cov}\left(P,Q\right)\\ \mathrm{cov}\left(Q,R\right)\\ \mathrm{cov}\left(P,R\right) \end{array}\right],V_{P}=\left[\begin{array}{c} \mathrm{var}\left(\;{\hat{\theta }}_{1}\right)\\ \mathrm{var}\left(\;{\hat{\theta }}_{2}\right)\\ \mathrm{var}\left(\;{\hat{\theta }}_{3}\right)\\ \mathrm{cov}\left(\;{\hat{\theta }}_{1},\;{\hat{\theta }}_{2}\right)\\ \mathrm{cov}\left(\;{\hat{\theta }}_{2},\;{\hat{\theta }}_{3}\right)\\ \mathrm{cov}\left(\;{\hat{\theta }}_{1},\;{\hat{\theta }}_{3}\right) \end{array}\right]$$
and $\theta_{1},\theta_{2}$, and $\theta_{3}$ are the distribution parameters; *D* is the matrix with elements $d_{ij}$ (1≤i,j≤6), which are the partial derivatives of the moments with respect to the distribution parameters. For example:(9)$$\begin{array}{l} d_{11}={\left(\partial P/\partial \theta_{1}\right)}^{2},\;\;\;\;\;d_{12}={\left(\partial P/\partial \theta_{2}\right)}^{2},\;\;\;\;\;d_{13}={\left(\partial P/\theta_{3}\right)}^{2},\;\;\;\;\\ \;d_{14}=2\left(\partial P/\partial \theta_{1}\right)\left(\partial P/\partial \theta_{2}\right),\;\;\;\;\;d_{15}=\left(\partial P/\partial \theta_{2}\right)\left(\partial P/\theta_{3}\right),\;\;\;\;\;d_{16}=2\left(\partial P/\partial \theta_{1}\right)\left(\partial P/\partial \theta_{3}\right)\;,\ldots \ldots \; \end{array}$$

Consequently, the $V_{p}$ can be calculated using Equation (10) as long as the elements of *D* and the $V_{M}$ have been calculated.
(10)$$V_{p}=\;D^{-1}\;V_{M}\;$$
where $D^{-1}$ is the inverse matrix of *D*.

Substituting the elements of $V_{p}$ and the partial derivatives of $x_{T}$ with respect to distribution parameters into Equation (3), one can obtain the variances of quantile estimators. The confidence interval of quantile estimators can then be calculated by using Equation (2).

The variances and covariances of MOM parameter estimates are calculated by using the relationship between the parameters and the population moments, which is relatively simple and understandable. However, the calculation of the second and higher order sample moments introduces sampling errors, which affects the accuracy of the estimation. The ML method is frequently applied owing to its large sample properties of yielding consistent estimates with minimum variance. Estimates for small samples have found general acceptance in practice as well [28]. However, this method involves some complicated calculations and approximations, which makes it inconvenient. The POME requires less artificial assumption due to insufficient data. Though it is comparable to the ML in parameter estimation, POME has the advantages of simple and fast calculation [17]. The calculation POME-based confidence interval also requires some approximations. Therefore, it is necessary to compare the performance of different methods to choose the most efficient one.

## 3. Asymptotic Variances of Quantile Estimators for Different Distributions 

Three commonly used distributions—gamma distribution, P3 distribution, and EV1 were considered in this study. 

### 3.1. Gamma Distribution

The pdf of the gamma distribution is given by:(11)$$f(x)=\frac{1}{\alpha^{\beta }\Gamma \left(\beta \right)}x^{\beta -1}\exp\left(-x/\alpha \right)$$
where α and β are the scale and shape parameters, and Γ(⋅) is the gamma function, and 0<x<∞.

For the gamma distribution, the *T*-year quantile is given by:(12)$${\hat{x}}_{T}=\hat{\alpha }\hat{\beta }+K_{T}\sqrt{{\hat{\alpha }}^{2}\hat{\beta }}$$

Differentiation of Equation (12) with respect to α and β yields:(13)$$\frac{\partial x_{T}}{\partial \alpha }=\beta +K_{T}\sqrt{\beta }\frac{\alpha }{\left|\alpha \right|}$$
(14)$$\frac{\partial x_{T}}{\partial \beta }=\alpha +\frac{K_{T}}{2}\sqrt{\alpha^{2}/\beta }-\frac{\sqrt{\alpha^{2}}}{\beta }\frac{\partial K_{T}}{\partial C_{S}}$$
where $\partial K_{T}/\partial C_{S}$ can be calculated by using Wilson–Hilferty transformation [1].

#### 3.1.1. Estimation of Asymptotic Variances by MOM and ML

Based on MOM, the standard error of ${\hat{x}}_{T}$ for the gamma distribution can be calculated directly by [29]:(15)$$s_{T}^{2}=\frac{\mu_{2}}{n}\left[{\left(1+K_{T}C_{v}\right)}^{2}+\frac{1}{2}{\left(K_{T}+2C_{v}\frac{\partial K_{T}}{\partial \gamma_{1}}\right)}^{2}\left(1+C_{v}^{2}\right)\right]$$
where $C_{v}$ is the coefficient of variation, and $C_{v}=\mu_{2}^{1/2}/\mu_{1}^{\prime }$; $\gamma_{1}=C_{s}$ is the coefficient of skewness.

The asymptotic variance and covariances of ML parameter estimators are derived as [1]:(16)$$\left[\begin{array}{cc} \mathrm{var}\left(\alpha \right) & \mathrm{cov}\left(\alpha ,\beta \right)\\ & \mathrm{var}\left(\beta \right) \end{array}\right]=\left[\begin{array}{cc} \alpha^{2}\psi^{\prime }D & -\alpha D\\ & \beta D \end{array}\right]$$
where $\psi^{\prime }=\psi^{\prime }\left(\beta \right)\;=\frac{d^{2}\log\Gamma \left(\beta \right)}{d\beta^{2}}$ is the tri-gamma function; $D=\frac{1}{\beta \;\psi^{\prime }-1}$.

Substituting Equations (13) and (14) and the variances and covariances terms in Equation (16) into Equation (3) yields the variance of the ML quantile estimator.

#### 3.1.2. Estimation of Asymptotic Variances by POME

For the gamma distribution, the relation between parameters and constraints can be expressed as [17]:(17)$$\left\{ \begin{array}{l} E\left(x\right)=\alpha \beta \\ E\left[\ln\left(x\right)\right]=\ln\left(\alpha \right)+\psi \end{array}\right.$$
where $\psi =\psi \left(\beta \right)\;=\frac{d\log\Gamma \left(\beta \right)}{d\beta }$ is digamma function. The parameter estimators $\hat{\alpha }$ and $\hat{\beta }$ can be obtained by solving the following equations: (18)$$\left\{ \begin{array}{l} \alpha \beta =\bar{X}\\ \ln\left(\alpha \right)+\psi \left(\beta \right)=\bar{W} \end{array}\right.$$
where $\bar{X}$ is the sample mean of *x*, and $\bar{W}$ is the sample mean of the random variable W=ln(x).

Then, $V_{M}$ and $V_{p}$ are written by:(19)$$V_{M}=\left[\begin{array}{c} \mathrm{var}\left(\bar{X}\right)\\ \mathrm{var}\left(\bar{W}\right)\\ \mathrm{cov}\left(\bar{X},\bar{W}\right) \end{array}\right]\;,V_{p}=\left[\begin{array}{c} \mathrm{var}\left(\hat{\alpha }\right)\\ \;\mathrm{var}\left(\hat{\beta }\right)\\ \mathrm{cov}\left(\hat{\alpha },\hat{\beta }\right) \end{array}\right]\;$$

For the gamma distribution,
(20)$$\mathrm{var}\left(\bar{X}\right)=\mathrm{var}\left(x\right)/n=\alpha^{2}\beta /n$$

Exact formulas for computing $\mathrm{var}\left(\bar{W}\right)$ and $\mathrm{cov}\left(\bar{X},\bar{W}\right)$ of $V_{M}$ are derived in Appendix A. Accordingly,
(21)$$\mathrm{var}\left(\bar{W}\right)=\mathrm{var}\left(W\right)/n=\psi^{\prime }/n$$
(22)$$\mathrm{cov}\left(\bar{X},\bar{W}\right)=\alpha \;/n$$

Consequently, one obtains:(23)$$V_{M}=\frac{1}{n}\left[\begin{array}{c} \alpha^{2}\beta \\ \psi^{\prime }\\ \alpha \end{array}\right]$$

Additionally, taking the partial derivates of $\bar{X}$ and $\bar{W}$ with respect to α and β, one can obtain the matrix *D*:(24)$$D=\left[\begin{array}{ccc} \beta^{2} & \alpha^{2} & 2\alpha \beta \\ 1/\alpha^{2} & {\psi^{\prime }}^{2} & 2\psi^{\prime }/\alpha \\ \beta /\alpha & \alpha \psi^{\prime } & 1+\beta \psi^{\prime } \end{array}\right]$$

Thus, all the components of $V_{M}$ and *D* are obtained. Substituting $V_{M}$ and *D* [Equations (23) and (24)] into Equation (10) yields $V_{P}$. The variance of the quantile estimator can then be obtained by substituting the terms of $V_{P}$ and Equations (13) and (14) into Equation (3).

### 3.2. Pearson Type 3 (P3) Distribution 

The pdf of P3 distribution is given by:(25)$$f\left(x|\alpha ,\beta ,\gamma \right)=\frac{1}{\alpha \Gamma \left(\beta \right)}{\left(\frac{x-\gamma }{\alpha }\right)}^{\beta -1}e^{-\frac{x-\gamma }{\alpha }}\;,\gamma <x<\infty$$
where α,β and  γ are the scale, shape, and location parameters, respectively, and γ<x<∞.

The *T*-year quantile of P3 distribution is given by:(26)$${\hat{x}}_{T}=\hat{\alpha }\hat{\beta }+\hat{\gamma }+K_{T}\sqrt{{\hat{\alpha }}^{2}\hat{\beta }}$$

Taking partial derivatives of Equation (34) with respect to α,β,γ yields:(27)$$\frac{\partial x_{T}}{\partial \alpha }=\beta +K_{T}\sqrt{\beta }\frac{\alpha }{\left|\alpha \right|}$$
(28)$$\frac{\partial x_{T}}{\partial \beta }=\alpha +\frac{K_{T}}{2}\sqrt{\frac{\alpha^{2}}{\beta }}-\frac{\sqrt{\alpha^{2}}}{\beta }\frac{\partial K_{T}}{\partial C_{S}}$$
(29)$$\frac{\partial x_{T}}{\partial \gamma }=1$$

#### 3.2.1. Estimation of Asymptotic Variances by MOM and ML

For the P3 distribution, the asymptotic variance of MOM quantile estimator is given by:(30)$$s_{T}^{2}=\frac{\mu_{2}}{n}\left[1+\gamma_{1}K_{T}+\frac{K_{T}^{2}}{2}\left(\frac{3}{4}\gamma_{1}^{2}+1\right)+3K_{T}\frac{\partial K_{T}}{\partial \gamma_{1}}\left(\gamma_{1}+\frac{\gamma_{1}^{3}}{4}\right)+3{\left(\frac{\partial K_{T}}{\partial \gamma_{1}}\right)}^{2}\left(2+3\gamma_{1}^{2}+\frac{5\gamma_{1}^{4}}{8}\right)\right]$$

The asymptotic variance and covariances of ML parameter estimators are given by [1]:(31)$$\begin{array}{l} \left[\begin{array}{ccc} \mathrm{var}\left(\alpha \right) & \mathrm{cov}\left(\alpha ,\beta \right) & \mathrm{cov}\left(\alpha ,\gamma \right)\\ & \mathrm{var}\left(\beta \right) & \mathrm{cov}\left(\beta ,\gamma \right)\\ & & \mathrm{var}\left(\gamma \right) \end{array}\right]\\ =\left[\begin{array}{ccc} \frac{1}{n\alpha^{2}G}\left[\frac{\psi^{\prime }\left(\beta \right)}{\left(\beta -2\right)}-\frac{1}{{\left(\beta -1\right)}^{2}}\right] & -\frac{1}{n\alpha^{3}G}\left(\frac{1}{\beta -2}-\frac{1}{\beta -1}\right) & \frac{1}{n\alpha^{2}G}\left[\frac{1}{\beta -1}-\psi^{\prime }\left(\beta \right)\right]\\ & \frac{2}{\alpha^{2}\left(\beta -2\right)} & -\frac{1}{n\alpha^{3}G}\;\left[\frac{\beta }{\left(\beta -1\right)}-1\right]\\ & & \frac{1}{n\alpha^{2}G}\left[\beta \psi^{\prime }\left(\beta \right)-1\right] \end{array}\right] \end{array}$$
where $G=\frac{1}{\left(\beta -2\right)\alpha^{4}}\left[2\psi^{\prime }-\frac{2\beta -3}{{\left(\beta -1\right)}^{2}}\right]$.

Substituting Equations (27)–(29) and the variance and covariance terms in Equation (31) into Equation (3) yields the asymptotic variance of the quantile estimator.

#### 3.2.2. Estimation of Asymptotic Variances by POME

On the basis of POME, the relation between parameters and constraints for P3 distribution is given by [17]: (32)$$\left\{ \begin{array}{l} E\left(x\right)=\alpha \beta +\gamma \\ E\left[\ln\left(x-\gamma \right)\right]=\ln\left(\alpha \right)+\psi \\ \mathrm{var}\left(x\right)=\alpha^{2}\beta \end{array}\right.$$
where ψ=ψ(β)  is digamma function. The parameter estimators $\hat{\alpha }$, $\hat{\beta }$, and $\hat{\gamma }$ can be obtained by solving the following equations:(33)$$\left\{ \begin{array}{l} \alpha \beta +\gamma =\bar{X}\\ \ln\left(\alpha \right)+\psi \left(\beta \right)={\bar{W}}_{1}\\ \alpha^{2}\beta =S^{2} \end{array}\right.$$
where $\bar{X}$ and $S^{2}$ are the sample mean and variance of *x*, and ${\bar{W}}_{1}$ is the sample mean of the random variable $W_{1}$ defined as $W_{1}=\ln\left(x-\gamma \right)$.

Therefore, $V_{M}$ and $V_{p}$ can be written by:(34)$$V_{M}=\left[\begin{array}{c} \mathrm{var}\left(\bar{X}\right)\\ \mathrm{var}\left(S^{2}\right)\\ \mathrm{var}\left({\bar{W}}_{1}\right)\\ \mathrm{cov}\left(\bar{X},S^{2}\right)\\ \mathrm{cov}\left(S^{2},{\bar{W}}_{1}\right)\\ \mathrm{cov}\left(\bar{X},{\bar{W}}_{1}\right) \end{array}\right],V_{p}=\left[\begin{array}{c} \mathrm{var}\left(\hat{\alpha }\right)\\ \mathrm{var}\left(\hat{\beta }\right)\\ \mathrm{var}\left(\hat{\gamma }\right)\\ \mathrm{cov}\left(\hat{\alpha },\hat{\beta }\right)\\ \mathrm{cov}\left(\hat{\beta },\hat{\gamma }\right)\\ \mathrm{cov}\left(\hat{\alpha },\hat{\gamma }\right) \end{array}\right]$$

Following Phine [24], the $V_{M}$ is given by:(35)$$V_{M}=\frac{1}{n}\left[\begin{array}{c} \alpha^{2}\beta \\ 2\alpha^{4}\;\beta \left(\beta +3\right)\\ \psi^{\prime }\left(\beta \right)\\ 2\alpha^{3}\;\beta \\ \alpha^{2}\\ \alpha \end{array}\right]$$

Taking partial derivatives of $\bar{X},\;\bar{W}\;\;and\;S^{2}$ with respect to α, β, and γ yields the matrix *D*:(36)$$D=\left[\begin{array}{cccccc} \beta^{2} & \alpha^{2} & \;1 & 2\alpha \beta \; & 2\alpha & 2\beta \\ 4\alpha^{2}\beta^{2} & \alpha^{4}\; & \;0\; & 4\alpha^{3}\beta^{2} & 0 & 0\\ 1/\alpha^{2}\;\;\; & {\psi^{\prime }}^{2} & \;0\; & \;2\psi^{\prime }/\alpha \; & 0 & 0\\ 2\alpha \beta & \alpha^{3}\;\; & \;0\; & 3\alpha^{2}\beta & \;\;\alpha^{2} & 2\alpha \beta \\ 2\beta & \alpha^{2}\psi^{\prime } & \;0\; & \alpha \left(1+2\beta \psi^{\prime }\;\right) & 0 & 0\\ \beta /\alpha \; & \;\alpha \psi^{\prime }\; & \;0\; & 1+\beta \psi^{\prime }\; & \psi^{\prime } & 1/\alpha \end{array}\right]$$
where $\psi^{\prime }\;=\psi^{\prime }(\beta )\;=\frac{d^{2}\log\Gamma (\beta )}{d\beta^{2}}$ is the tri-gamma function.

Substituting $V_{M}$ and *D* [Equations (35) and (36)] into Equation (10) yields $V_{P}$. Accordingly, the asymptotic variance of the quantile estimator can then be obtained by substituting the terms of $V_{P}$ and Equations (27)–(29) into Equation (3). 

### 3.3. Extreme Value Type 1 (EV1) Distribution

The pdf and the cumulative distribution function of EV1 distribution can be expressed respectively as:(37)$$f\left(x\right)=\frac{1}{\alpha }\exp\left[-\frac{x-u}{\alpha }-\exp\left(-\frac{x-u}{\alpha }\right)\right]$$
(38)$$F\left(x\right)=\exp\left[-\exp\left(-\frac{x-u}{\alpha }\right)\right]$$
where α and u are the scale and shape parameters, respectively, and −∞<x<∞.

The *T*-year quantile of EV1 distribution can be obtained from Equation (38) by substituting F(x)=1−1/T and solving for *x*:(39)$${\hat{x}}_{T}=\hat{u}-\hat{\alpha }\log\left(-\log\left(1-1/T\right)\right)$$

Differentiating Equation (39) with α and *u* yields the derivatives of $x_{T}$ with respect to α and *u*:(40)$$\frac{\partial x_{T}}{\partial \alpha }=-\log\left(-\log\left(1-1/T\right)\right)$$
(41)$$\frac{\partial x_{T}}{\partial u}=1$$

#### 3.3.1. Estimation of asymptotic variance by MOM and ML 

The asymptotic variance of MOM quantile estimator is given by:(42)$$s_{T}^{2}=\frac{\mu_{2}}{n}\left(1+1.1396K_{T}+1.1K_{T}^{2}\right)$$
where $K_{T}$ is given by:(43)$$K_{T}=-\frac{\sqrt{6}}{\pi }\left[\log\left(-\log\left(1-\frac{1}{T}\right)\right)+\varepsilon \right]$$
where ε is Euler constant, ε=0.5772157.

The variance and covariances for the ML parameter estimators are given by [1]:(44)$$\left[\begin{array}{cc} \mathrm{var}\left(\alpha \right) & \mathrm{cov}\left(\alpha ,u\right)\\ & \mathrm{var}\left(u\right) \end{array}\right]=\frac{\alpha^{2}}{n}\left[\begin{array}{cc} 0.8046 & 0.2287\\ & 1.1128 \end{array}\right]$$

Substituting Equations (40) and (41) and the variance and covariance terms in Equation (44) into Equation (3) yields the asymptotic variance of the quantile estimator.

#### 3.3.2. Estimation of Asymptotic Variances by POME

The relation between parameters and constraints for EV1 distribution can be expressed as [17]:(45)$$\left\{ \begin{array}{l} E\left[\frac{\left(x-u\right)}{\alpha }\right]=\varepsilon \\ E\left[\exp\left(-\frac{x-u}{\alpha }\right)\right]=1 \end{array}\right.$$

The estimators $\hat{\alpha }$ and $\hat{u}$ of the parameters can be obtained by solving the following equations:(46)$$\left\{ \begin{array}{l} \bar{Y}=\varepsilon \\ \bar{V}=1 \end{array}\right.$$
where $\bar{Y}$ and $\bar{V}$ are the sample mean of variables defined by y=(x−u)/α and V=exp(−y), respectively.

The variances and covariances of the moments and parameter estimators are written respectively as:(47)$$V_{M}=\left[\begin{array}{c} \mathrm{var}\left(\bar{Y}\right)\\ \mathrm{var}\left(\bar{V}\right)\\ \mathrm{cov}\left(\bar{Y},\bar{V}\right) \end{array}\right],V_{p}=\left[\begin{array}{c} \mathrm{var}\left(\hat{\alpha }\right)\\ \mathrm{var}\left(\hat{u}\right)\\ \mathrm{cov}\left(\hat{\alpha },\hat{u}\right) \end{array}\right]$$

According the derivations in [17], the $V_{M}$ is given by: (48)$$V_{M}=\frac{1}{n}\left[\begin{array}{c} \pi^{2}/6\\ 1\\ -1 \end{array}\right]$$

Taking partial derivatives of $\bar{Y}$ and $\bar{V}$ with respect to α and u yields:(49)$$D=\alpha^{-2}\left[\begin{array}{ccc} \varepsilon^{2} & 1 & 2\varepsilon \\ {\bar{W}}^{2} & 1 & 2\bar{W}\\ -\varepsilon \bar{W} & -1 & -\left(\varepsilon +\bar{W}\right) \end{array}\right]$$
where W=yexp(−y).

Substituting $V_{M}$ and *D* [Equations (48) and (49)] into Equation (10) yields $V_{P}$. The asymptotic variance of quantile estimator can then be obtained by substituting the terms of $V_{P}$ and Equations (40) and (41) into Equation (3).

## 4. Simulation Experiments

In this section, the Monte Carlo simulation experiments were performed to evaluate the performance of POME in calculation of the asymptotic variances and confidence intervals of quantiles and to compare it with the MOM and ML methods. In this study, four kinds of data sets were generated from the Wakeby distribution with parameters as shown in Table 1 [16,30]. The quantile function of the Wakeby distribution is given by [31]:(50)$$x\left(F\right)=\xi +\frac{\alpha }{\beta }\left[1-{\left(1-F\right)}^{\beta }\right]-\frac{\gamma }{\delta }\left[1-{\left(1-F\right)}^{-\delta }\right]$$
where F is the uniform (0,1) variate, and ξ,   α,   β,   γ,  δ are the parameters.

*N*s = 1000 samples with size *n* (*n* = 20, 50, 100, 1000) were generated from each Wakeby distribution. Then, the quantiles ${\hat{x}}_{T}$ corresponding to different return periods (*T* = 10, 100, and 200) and their asymptotic variances and confidence intervals were calculated for EV1. Table 2 lists the median values of the estimated quantiles (${\hat{x}}_{T}$), standard errors (St), and confidence interval width (CI width).

From Table 2, generally for all methods and for all cases, it was observed that the standard errors and confidence interval widths of the quantiles increased with the return period *T* and decreased with the sample size. For all cases, the selected three methods exhibited very similar behaviors. Thus, we would take case III as an example in the latter discussion. 

From case III, it was observed that the POME generally gave the smallest median of both standard errors and confidence interval widths of quantiles regardless of the sample size and return period. MOM was always the worst of the three competing methods and gave the largest results. The results of ML fell between MOM and POME. For example, when the sample size equaled 50, the median standard errors of MOM, ML, and POME quantile estimator for return period *T* = 100 were 69.2, 66.2, and 63.5, respectively. Correspondingly, the confidence interval widths were 271.4, 259.4, and 248.8, respectively, which indicated that the uncertainty of the POME estimator was less than that of the MOM and the ML estimators. Therefore, the performance of the POME was found to be superior to the MOM and the ML.

In addition, for each method considered, the median of both standard errors and confidence interval widths of quantiles decreased significantly when the sample size increased from 20 to 1000. For *T* = 100, when the sample size increased from 20 to 1000, the median of standard errors of MOM quantile estimators decreased from 109.5 to 8.2, the median of standard errors of ML quantile estimators decreased from 99.9 to 8.7, and the median of standard errors of POME quantile estimators decreased from 99 to 8.3. The median of confidence interval widths decreased from 429.3 to 32.2 for MOM quantile estimators, 391.8 to 34.2 for ML quantile estimators, and 388 to 32.6 for POME quantile estimators. This was an indication of the influence sample size had on the estimation accuracy.

## 5. Application 

The annual precipitation data from four gauging stations at the Weihe River basin in China were considered as the case study. All data were obtained from the National Climate of China Meteorological Administration and were complete. The detailed information of these data is given in Table 3.

The gamma, P3, and EV1 distributions were used to fit the data set, and the MOM, ML, and POME were used to estimate the parameters of these distributions, as given in Table 4. It can be seen that the parameters of the gamma distribution estimated by MOM, ML, and POME were very close, as were the EV1 distribution, while those of the P3 distribution departed significantly.

To evaluate and compare the performances of the three methods and the distributions, the ordinary least square (OLS) criterion, Akaike information criterion (AIC), and quasi-optimal deterministic coefficient test (QD) were employed and can be defined as:(51)$$OLS=\sqrt{\frac{1}{n}\sum_{i=1}^{n}{\left(x_{i}-{\hat{x}}_{i}\right)}^{2}}$$
(52)$$AIC=n\ln\left(\frac{1}{n}\sum_{i=1}^{n}{\left(x_{i}-{\hat{x}}_{i}\right)}^{2}\right)+2m$$
(53)$$QD=1-\frac{\sum_{i=1}^{n}{\left(x_{i}-{\hat{x}}_{i}\right)}^{2}}{\sum_{i=1}^{n}{\left(x_{i}-\bar{x}\right)}^{2}}$$
where $x_{i}$ and ${\hat{x}}_{i}$ are the observed data and the predicted values of a given (*i*-th) quantile, respectively, $\bar{x}$ is the mean value of observed data, *m* is the number of parameters of a given model, and *n* is the sample size.

The OLS criterion is recommended as a curve optimization rule for measuring the difference between empirical and theoretical values in hydrological frequency analysis in China. The smaller OLS values represent the better performance of the model. The AIC is more appropriate for the comparison of models have different number of parameters. Given a set of candidate models for the data, the best model is the one with the minimum AIC value. QD is used to describe the fitting degree of observed values and theoretical values and the best fit model is the one that gets the QD value closest to 1. The OLS, AIC, and QD were calculated as given in Table 5. 

It is seen from Table 5 that the selected best parameter estimation method for each distribution by the three criterions is coincident and the result of the best fitted distribution for each station by the three criterions is the same as well. Take the Changwu station in Table 5 for example. According to the smallest OLS and AIC values and the largest QD values, the POME, MOM and POME are suggested to be the best methods for parameter estimation for Gamma, P3 and EV1 distributions, respectively. And the best fitted distribution for Changwu station recommended by the OLS, AIC and QD criteria is P3 distribution. Additionally, according to the results given in Table 5, the best fitted distributions for the gauging stations Meixian, Tongguan and Lintong recommended by the OLS, AIC and QD methods, is EV1 distribution with the parameters estimated by POME. Thus the best estimation method for each station is POME and this is coincident with the results of the simulation experiments in Section 4, which shows that the performance of POME is better than MOM and ML. The bold values in the table denote the smallest OLS and AIC values and the largest QD values.

The quantiles along with the standard errors and 95% confidence intervals for 10, 20, 50, 100, 200, and 500 years return periods of the best fitted distribution based on the parameters estimated by POME are given in Table 5. For the sake of comparison, the quantiles, standard errors, and 95% confidence interval widths based on MOM and ML are also given in Table 6. The results show that the standard errors and confidence interval widths of quantile estimators obtained by POME were smaller than those obtained by the MOM and the ML methods with the exception of the results of *T* = 10 at Changwu station, which indicated that the POME yielded more precise parameters and quantiles estimations. 

To better understand the performance of the different methods, the differences in the uncertainty reductions for the standard errors and 95% confidence interval widths of the quantile estimators were given in terms of relative deviation, as shown in Table 7. For the relatively long return period (T≥50), there were significant reductions in the standard errors and 95% confidence interval widths obtained by POME compared to MOM. For example, for a return period of *T* = 500, the reductions in standard errors and the confidence interval widths were of about 32%, 17%, 17%, and 16% for Changwu, Lintong, Meixian, and Tongguan, respectively.

It can also be seen from Table 6 that, for Changwu station, the reduction in standard errors and 95% confidence interval widths obtained by POME were significant when compared to ML. For example, the reductions in the standard errors and confidence interval widths of a 500-year quantile was about 19%. For Lintong, Meixian, and Tongguan stations, the reductions were relatively smaller—about 6%, 6%, and 5%, respectively. Overall, the POME provided more accurate quantile estimators.

## 6. Conclusions

In this study, the POME method was applied for the estimation of the asymptotic variances and confidence intervals of quantiles, and the corresponding calculation formulas for gamma, P3, and EV1 distributions based on the POME method were deduced. The calculation procedures of the MOM and the ML methods were also reviewed briefly for comparison. The Monte Carlo simulation experiments were carried out to evaluate the performance of the POME method and to compare it with the MOM and the ML methods. In addition, annual precipitation data from four stations at the Weihe River basin in China were selected as the case study. The following conclusions were drawn from this study: (1)The calculation formulas of the asymptotic variances and confidence intervals of quantiles for three distributions based on POME are given. The results of simulation experiments and the case study show that the POME method can provide an effective way for reducing the uncertainty of quantile estimators.(2)Results of the simulation experiments demonstrate that the POME method yields the smallest standard errors and the narrowest confidence intervals of quantile estimators compared with the results of MOM and ML. This may benefit from fewer sampling errors and approximation in derivation. Thus, the POME can give more accurate estimates. Furthermore, the standard errors and confidence interval widths of the quantiles increased with the return period *T* and decreased with the sample size.(3)Results of the case study indicate that when using different criteria for distribution selection, the results are coincidental, and the POME is the optimal method for parameter estimation. Furthermore, the POME can give more reliable precipitation quantiles since the standard errors and 95% confidence interval widths of precipitation quantiles obtained by POME are smaller than those obtained by the MOM and the ML methods.

This study investigated the calculation of asymptotic variances and confidence intervals based on POME for three commonly used distributions and compared the performance of POME with that of MOM and ML. In addition, the POME-based asymptotic variances and confidence intervals of quantiles for more distributions deserve more thorough investigation.

## Figures and Tables

**Table 1 entropy-21-00315-t001:** The Monte Carlo simulation data sets generated from the Wakeby distribution.

Case	*ξ*	*α*	*β*	*γ*	*δ*	*C*v	*C*s
I	0	16	16	0.4	0.04	0.38	1.10
I	15.4	308.8	10.25	38.5	−0.30	0.36	0.48
III	273.69	521.10	1.25	4.77	−0.21	0.24	0.64

**Table 2 entropy-21-00315-t002:** Median of estimated quantiles (${\hat{x}}_{T}$), standard error (St), and 95% confidence interval (CI) width from generated data; MOM = methods of moments, ML = maximum likelihood, POME = principle of maximum entropy.

Case	Sample Size *n*	Return Period *T*	MOM			ML			POME		
			${\hat{x}}_{T}$	St	CI Width	${\hat{x}}_{T}$	St	CI Width	${\hat{x}}_{T}$	St	CI Width
**I**	**20**	10	1.97	0.22	0.85	2.08	0.22	0.85	2.02	0.20	0.80
		100	2.83	0.41	1.60	3.06	0.38	1.49	2.97	0.35	1.37
		200	3.09	0.47	1.83	3.36	0.43	1.69	3.26	0.40	1.55
	50	10	2.00	0.14	0.56	2.13	0.15	0.58	2.06	0.13	0.52
		100	2.89	0.27	1.05	3.20	0.26	1.01	3.05	0.23	0.89
		200	3.14	0.31	1.20	3.52	0.29	1.14	3.35	0.26	1.00
	100	10	2.01	0.10	0.40	2.16	0.11	0.42	2.08	0.10	0.37
		100	2.93	0.19	0.76	3.24	0.19	0.73	3.09	0.16	0.64
		200	3.20	0.22	0.87	3.56	0.21	0.82	3.39	0.18	0.72
	1000	10	2.02	0.03	0.13	2.17	0.03	0.13	2.09	0.03	0.12
		100	2.95	0.06	0.24	3.27	0.06	0.23	3.12	0.05	0.20
		200	3.23	0.07	0.28	3.60	0.07	0.27	3.42	0.06	0.23
**II**	20	10	106.9	12.1	47.3	111.1	11.9	46.5	109.7	11.4	44.7
		100	155.3	22.8	89.3	164.7	20.7	81.3	161.3	19.7	77.2
		200	169.8	26.0	102.1	180.4	23.4	91.9	176.8	22.2	87.1
	50	10	106.9	12.1	47.3	112.3	7.7	30.2	110.3	7.4	28.9
		100	155.3	22.8	89.3	167.8	13.5	52.8	163.7	12.8	50.0
		200	169.8	26.0	102.1	184.4	15.2	59.7	179.4	14.4	56.5
	100	10	107.1	5.5	21.5	113.0	5.5	21.6	110.8	5.3	20.6
		100	156.3	10.3	40.5	169.0	9.6	37.7	164.8	9.1	35.7
		200	170.9	11.8	46.4	185.6	10.9	42.6	180.8	10.3	40.3
	1000	10	107.5	1.7	6.8	113.5	1.8	6.9	111.2	1.7	6.6
		100	156.7	3.3	12.9	169.9	3.1	12.0	165.5	2.9	11.3
		200	171.3	3.8	14.7	186.6	3.5	13.6	181.6	3.3	12.8
**III**	20	10	676.4	58.1	227.6	685.7	57.2	224.1	698.5	57.2	224.4
		100	905.4	109.5	429.3	944.9	99.9	391.8	958.1	99.0	388.0
		200	974.0	125.2	490.9	1023.0	113.0	442.9	1036.1	111.6	437.6
	50	10	675.0	36.7	143.9	700.5	37.8	148.4	700.2	36.8	144.3
		100	905.5	69.2	271.4	971.7	66.2	259.4	965.6	63.5	248.8
		200	974.3	79.2	310.3	1053.7	74.8	293.3	1044.3	71.7	280.9
	100	10	1065.5	92.4	362.0	1160.3	86.3	338.2	1148.5	82.5	323.5
		100	674.8	26.0	102.0	707.7	65.5	106.9	700.2	26.2	102.7
		200	906.9	49.1	192.4	985.3	41.5	186.9	968.7	45.2	177.4
	1000	10	975.7	56.1	220.0	1067.2	47.7	211.3	1048.4	51.1	200.2
		100	674.7	8.2	32.2	713.6	8.7	34.2	701.4	8.3	32.6
		200	906.9	15.5	60.8	994.1	15.3	59.8	971.3	14.4	56.3

**Table 3 entropy-21-00315-t003:** Basic information on each of the monthly precipitation series used in this study.

Station Name	Record Length (Year)	Mean (mm)	Standard Deviation	Coefficient of Variation	Skewness	First-Order Serial Correlation Coefficient
Changwu	51	580.6	131.8177	0.2270	0.5070	3.2153
Lintong	50	579.5	129.2014	0.2230	0.6299	3.7670
Meixian	50	578.0	129.7214	0.2245	0.5828	3.4614
Tongguan	52	605.5	143.4648	0.2369	0.5771	3.6438

**Table 4 entropy-21-00315-t004:** Parameter values of each distribution estimated by the three methods; P3 = Pearson type 3, EV1 = extreme value type 1.

Station Name	Method	Gamma		P3		EV1		
		*α*	*β*	*α*	*β*	*γ*	*α*	*u*
**Changwu**	MOM	29.9282	19.3993	15.5623	33.4147	60.5782	102.7783	521.2619
	ML	29.0510	19.9851	16.0097	32.5878	58.8660	114.9989	518.3959
	POME	29.3458	19.7893	11.1017	39.5620	141.3794	111.0142	516.5091
**Lintong**	MOM	28.8063	20.1169	10.0804	40.6938	169.2817	100.7383	521.3449
	ML	27.6873	20.9299	17.1488	30.6908	53.1797	113.0664	519.0634
	POME	28.6849	20.2192	10.3139	40.2305	164.5577	108.5086	516.8608
**Meixian**	MOM	29.1161	19.8498	11.7762	37.8015	132.7913	101.1438	519.5689
	ML	28.0875	20.5768	17.7271	30.3383	40.1383	113.7362	517.1681
	POME	28.7575	20.1089	10.8002	39.4726	151.6378	109.1477	514.9499
**Tongguan**	MOM	33.9927	17.8122	12.0090	41.3991	108.3228	111.8595	540.9202
	ML	32.9535	18.3740	15.0343	36.5741	55.6191	125.0433	537.9242
	POME	33.7243	17.9653	10.3756	44.5388	143.3688	120.6552	535.8444

**Table 5 entropy-21-00315-t005:** Ordinary least square (OLS), Akaike information criterion (AIC) and quasi-optimal deterministic coefficient test (QD) values of three distributions calculated by MOM, ML, and POME.

Station Name	Method	Gamma			P3			EV1		
		OLS	AIC	QD	OLS	AIC	QD	OLS	AIC	QD
Changwu	MOM	16.6696	292.9861	0.9837	16.3741	291.1613	0.9843	19.3927	308.4197	0.9779
	ML	17.5392	298.1729	0.9819	17.0194	295.1041	0.983	16.8854	294.2978	0.9833
	POME	17.2128	296.2565	0.9826	**16.0997**	**289.4375**	**0.9848**	16.6415	292.8135	0.9837
Lintong	MOM	19.6949	304.0361	0.9763	18.516	297.8638	0.979	18.8765	299.792	0.9782
	ML	20.6502	308.7725	0.9739	20.1246	306.1944	0.9752	16.3975	285.7129	0.9836
	POME	19.7466	304.2982	0.9762	18.5442	298.0155	0.979	**16.0775**	**283.742**	**0.9842**
Meixian	MOM	17.9613	294.8219	0.9804	16.9804	289.2061	0.9825	17.8405	294.1469	0.9807
	ML	18.9359	300.106	0.9783	18.5121	297.8426	0.9792	15.0757	277.3083	0.9862
	POME	18.2446	296.3869	0.9798	16.8553	288.4664	0.9828	**14.6984**	**274.7736**	**0.9869**
Tongguan	MOM	20.419	319.7126	0.9793	20.0141	317.6294	0.9802	21.9677	327.3156	0.9761
	ML	21.2629	323.9242	0.9776	20.8105	321.6875	0.9785	19.5845	315.373	0.981
	POME	20.5825	320.542	0.979	19.9542	317.3178	0.9803	**19.2731**	**313.7059**	**0.9816**

Bold values indicate the smallest OLS and AIC values and the largest QD values.

**Table 6 entropy-21-00315-t006:** Quantile estimators, standard error, and 95% confidence interval widths based on MOM, ML, and POME for the annual precipitation (mm).

Station Name	Best fitted Distribution	Return Period (Year)	MOM			ML			POME		
			Quantile	Standard Error	Confidence Interval Width	Quantile	Standard Error	Confidence Interval width	Quantile	Standard Error	Confidence Interval Width
Changwu	P3	10	755.0	31.2	122.1	753.1	30.5	119.6	755.7	31.7	124.3
		20	814.7	40.7	159.7	811.9	38.9	152.3	817.5	38.4	150.7
		50	885.6	56.1	220.0	881.9	51.7	202.5	891.6	47.3	185.6
		100	935.3	69.1	270.7	930.8	62.2	243.8	943.8	54.0	211.6
		200	982.3	82.8	324.5	977.1	73.3	287.2	993.5	60.6	237.5
		500	1041.4	101.8	398.9	1035.3	88.5	347.0	1056.3	69.2	271.2
Lintong	EV1	10	704.4	31.8	124.7	724.5	32.2	126.1	714.0	30.6	120.0
		20	748.0	37.5	146.9	773.5	37.0	144.9	761.0	35.0	137.1
		50	820.6	47.4	185.7	854.9	45.2	177.3	839.2	42.5	166.7
		100	984.8	70.7	277.1	1039.2	64.6	253.3	1016.0	60.3	236.2
		200	1054.8	80.8	316.9	1117.8	73.1	286.4	1091.5	68.0	266.5
		500	1147.3	94.3	369.7	1221.6	84.3	330.3	1191.1	78.2	306.6
Meixian	EV1	10	703.3	31.9	125.2	723.8	32.4	126.9	713.3	30.8	120.7
		20	747.2	37.6	147.5	773.1	37.2	145.8	760.6	35.2	137.9
		50	820.0	47.6	186.4	855.0	45.5	178.3	839.1	42.8	167.6
		100	984.8	71.0	278.3	1040.4	65.0	254.8	1017.0	60.6	237.6
		200	1055.2	81.2	318.2	1119.5	73.5	288.1	1093.0	68.4	268.0
		500	1148.0	94.7	371.2	1223.9	84.8	332.3	1193.2	78.7	308.4
Tongguan	EV1	10	744.2	34.6	135.7	765.1	34.9	136.8	755.1	33.5	131.5
		20	792.6	40.8	160.0	819.3	40.1	157.1	807.4	38.4	150.4
		50	873.2	51.6	202.2	909.3	49.0	192.2	894.2	46.7	183.1
		100	1055.5	77.0	301.8	1113.1	70.1	274.7	1090.9	66.3	259.9
		200	1133.3	88.0	345.1	1200.1	79.2	310.6	1174.8	74.8	293.3
		500	1236.0	102.7	402.5	1314.9	91.4	358.2	1285.5	86.1	337.6

**Table 7 entropy-21-00315-t007:** Change in the uncertainty in quantile estimators based on POME compared with the MOM and ML methods (%).

Station Name	Return Period(Year)	POME to MOM			POME to ML		
		Quantile	Standard Error	Confidence Interval Width	Quantile	Standard Error	Confidence Interval Width
Changwu	10	0.09	1.74	1.74	0.34	3.78	3.78
	20	0.34	−5.63	−5.63	0.68	−1.02	−1.02
	50	0.67	−15.66	−15.66	1.10	−7.68	−7.68
	100	0.91	−21.83	−21.83	1.39	−11.88	−11.88
	200	1.14	−26.82	−26.82	1.67	−15.33	−15.33
	500	1.43	−32.02	−32.02	2.02	−18.99	−18.99
Lintong	10	1.37	−3.74	−3.74	−1.49	−4.93	−4.93
	20	1.74	−6.67	−6.67	−1.67	−5.29	−5.29
	50	2.27	−10.23	−10.23	−1.92	−5.70	−5.70
	100	3.17	−14.76	−14.76	−2.35	−6.18	−6.18
	200	3.48	−15.92	−15.92	−2.50	−6.29	−6.29
	500	3.82	−17.05	−17.05	−2.66	−6.41	−6.41
Meixian	10	1.41	−3.58	−3.58	−1.50	−4.95	−4.95
	20	1.79	−6.51	−6.51	−1.68	−5.32	−5.32
	50	2.34	−10.08	−10.08	−1.93	−5.73	−5.73
	100	3.27	−14.62	−14.62	−2.37	−6.21	−6.21
	200	3.58	−15.78	−15.78	−2.51	−6.33	−6.33
	500	3.93	−16.92	−16.92	−2.68	−6.44	−6.44
Tongguan	10	1.47	−3.14	−3.14	−1.35	−3.92	−3.92
	20	1.86	−5.98	−5.98	−1.51	−4.20	−4.20
	50	2.41	−9.45	−9.45	−1.73	−4.53	−4.53
	100	3.35	−13.88	−13.88	−2.11	−4.92	−4.92
	200	3.66	−15.00	−15.00	−2.23	−5.02	−5.02
	500	4.01	−16.12	−16.12	−2.37	−5.11	−5.11

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
