# Peer review of "Confidence Interval Estimation for Precipitation Quantiles Based on Principle of Maximum Entropy"

_entropy, 2019, doi:10.3390/e21030315_

Round 1
Reviewer 1 Report
The manuscript entitled “Confidence interval estimation for precipitation quantiles based on principle of maximum entropy” is scientifically sound and brings a major contribution to stochastic hydrology. The text is well written, the application is relevant, and the discussion and conclusions are adequate. The authors tackle a relevant issue, i.e., the uncertainty of extreme hydrological values, using the principle of maximum entropy. The mathematics of the paper, in which the authors derive the formulas of asymptotic variances, is robustly dealt with. The paper compares the performance of the principle of maximum entropy (POME) with those of the methods of moments and of the maximum likelihood, showing the superiority of the results using POME. Considering the merits of the manuscript, we understand that it can be accepted for publication in its present form, except for minor details.
Please check for the following words or sentences:
The word "of" (L 8)
“…corresponding to a specified confidence and including the true value” (L 9): we understand that the true value may be observed outside the confidence interval, please check
“It is shown that POME yields the smallest standard errors and the narrowest confidence intervals of quantile estimators among the three methods, and can reduce the uncertainty of quantile estimators” (LL 19-21): could you emit an opinion on why this happens (both in the discussion and in the conclusion/abstract)?
“Stedinger (1983), Ashkar and Bobee (1988)…” (L 58): check for the style of the Journal
Please write Weibull instead of weibull L 77
We suggest to use “the two-component…” in L 100
Define Cs (L 257)
Check for “choose” (L 438)
Where was the “standard error” (L 459) defined?
Table 2 caption: please mention the precipitation unit
Table 4 caption does not state clearly to which data the performance assessment refers to
We suggest that the authors use the unit (years) when referring to the return period
Table 6: if negative values mean reduction, the variable (see caption) should not be "reduction"
We suggest “the POME-based confidence intervals…” (L 555).
Author Response
Dear Editors and Reviewers,
We are very grateful for the valuable comments of the editor and reviewers of our manuscript (“Confidence interval estimation for precipitation quantiles based on principle of maximum entropy”, ID: entropy-436969 ). Those comments are all valuable and very helpful for revising and improving our paper. We have studied comments carefully and modified the manuscript accordingly. The detailed corrections are listed point by point in the attachment file. In the revised manuscript, the revised parts were marked by red font. Finally, this manuscript has been polished by a native English speaking colleague before submitted. If you have any question about this manuscript, please do not hesitate to contact us.
Yours sincerely,
Ting Wei, Songbai Song

Reviewer 2 Report
This is a detailed and clearly structured paper although I find it difficult to judge the merits of the research and whether it is suitable for publication in the international literature. In a number of places, and particularly in the Abstract, the standard of English is below that required, and would benefit from professional help. More generally, the paper is overlong, the wider rationale and motivation for the study is unclear, as is the justification for selecting the data-set used in the analysis, and the wider implications / transferability of the study. For these reasons, I don’t think the study, in its present form, is suitable for publication and the authors should be encouraged to substantially revise and resubmit their paper.
Author Response
Dear Editors and Reviewers,We are very grateful for the valuable comments of the editor and reviewers of our manuscript (“Confidence interval estimation for precipitation quantiles based on principle of maximum entropy”, ID: entropy-436969 ). Those comments are all valuable and very helpful for revising and improving our paper. We have studied comments carefully and modified the manuscript accordingly. The detailed corrections are listed point by point in the attachment file. In the revised manuscript, the revised parts were marked by red font. Finally, this manuscript has been polished by a native English speaking colleague before submitted. If you have any question about this manuscript, please do not hesitate to contact us.
Yours sincerely,
Ting Wei, Songbai Song

Reviewer 3 Report
The review of the article
Confidence interval estimation for precipitation quantiles
based on principle of maximum entropy
by
Ting Wei, Songbai Song
submitted to Entropy (ISSN 1099-4300)
entropy-436969
In their paper Authors examine the confidence interval of the principle of maximum entropy (POME) quantile estimators. In their study, the calculation formulas of asymptotic variances and confidence intervals of quantiles based on POME for Gamma, Pearson type 3 (P3) and Extreme value type 1 (EV1) distributions were derived. The theoretical calculations were confirmed for finite samples by the Monte Carlo Simulation experiments. Using four data sets for annual precipitation at the Weihe River basin in China, the derived formulas were applied for calculating the variances and confidence intervals of precipitation quantiles for different return periods and the results were compared with those of the methods of moments (MOM) and of maximum likelihood (ML) method. The Authors showed that POME yields the smallest standard errors and the narrowest confidence intervals of quantile estimators among the three methods, and can reduce the uncertainty of quantile estimators.
Overall remarks
I would like to thank the Authors for the interesting and inspiring article. I really enjoyed reading the paper, because it is well structured and comprehensive, though sometimes too wordy. On the other hand its novelty is either not exposed in a vivid way or just weak. Let me just concentrate on the critical issues concerning the article which you can find useful while preparing a new version.
The first chapter presents the thorough review of the literature on research concerning various aspects of accuracy in estimation of the hydrological parameters. However true and interesting, the literature cited seems to be a bit outdated. Is it possible that nothing happened in this field of knowledge for over 10 years? Besides, the Authors refer to the facts that are either trivial, well known or even fossilised in the hydrological literature (e.g. “Different estimation methods may yield different standard errors of the 33 estimated quantities, and thus the confidence intervals are different as well.”, p. 2, l. 32-33), so they might be as well omitted in the manuscript. The Authors state it clearly (p. 6, l. 130-135) the purpose of the paper, but it would be good if they underlined the innovative issues in their work.
In the second chapter “Confidence interval estimation of the quantile” the Authors present the theory and methodology of calculating asymptotic variance and confidence intervals for the quantiles of the T-long return period. The methodology and formulae concerning the MOM- and ML-estimated variances of quantiles presented in sections 2.1 up to 2.2.2 are the good reminders (easy to find in any statistical handbooks), but in my opinion for the clarity of the text they should be moved to the appendices and the Authors might then concentrate on the final formulae for these methods and POME estimators. The Authors should also carry out a-few-sentences comparison of these thee methods of estimation – their pros, cons and limitations, also in terms of their data requirements and calculation-time intensity, because the information about narrowness of the confidence intervals by themselves does not make one method superior over the other two.
In the third chapter the Authors describe the derivation of the formulae of the asymptotic standard error for three selected statistical models, namely for Gamma, Pearson type 3 and Extreme value type 1 (i.e. Gumbel) distribution functions. Indeed, these three models are often used for estimation of the upper (flood) quantiles the Flood Frequency Analysis, however I would like to know why the Authors selected these three distribution functions, especially that the Gamma PDF is a special case of the Pearson type 3 distribution function. Consequently, I would like to know also, whether is it possible to derive POME estimators of the confidence intervals for other commonly used distributions in the Flood Frequency Analysis (GEV, Weibull, Two-Component Extreme Value - TCEV, etc.). Again, here a few commonly known equations could be moved to the appendix.
In the fourth chapter the author describe the results of the Monte Carlo experiment comparing the theoretical confidence intervals with the ones calculated by means of the simulations for with known and finite datasets. The Authors assumed that the synthetic data represent the population of the 5-parameter specific Wakeby distribution with the selected parameters. A few issues in this chapter remain for me unclear. First of all I do not understand why the Authors concentrated only on one specific set of parameters of the Wakeby distribution, yielding a very small coefficient of skewness, CS = 0.00185 (with mean = 0.92, CV = 0.5). One can easily spot that the constant coefficient of skewness for Gumbel distribution is 600 times larger. For the sake of scientific curiosity, I would check what is the result for other specific Wekeby distribution sets of parameters (see Strupczewski et al 2005, Kochanek, et al 2005) and for the True=True case, i.e. when the sample is generated by the distribution used for estimation of quantiles. For such conditions the results may lead for completely different conclusions. It is interesting also why the Authors limited their calculations to the EV1 distribution only and to the sample size up to 80 elements, far from N ® ¥. I would suggest to perform the calculations for e.g. N = 1000 elements to see whether the asymptotic formulae agrees with the simulations (see Kochanek, et al 2005).
The fifth chapter “Application” considers the case study for the River Weihe Basin in China. The data consider four selected water gauging stations. The figures in the Table 2 clearly indicate that the hydrological conditions (mean, CV and CS) in the River Weihe are far from the one applied in the specific Wakeby in fourth chapter. Maybe it would be good to adjust the conditions in the simulation to the real hydrological regime of the River Weihe to meke the results more comparable?
The sixth chapter concludes the paper. The last section is, in my opinion, a summary of the previous chapters rather than the conclusions drawn from the research. The bullet-point form of the conclusions is fine, however it would be good if the Authors draw any take-home conclusions or recommendations of general character.
Summary and recommendations
In my opinion the reviewed paper needs some minor work to meet the standards indispensable to be published in the Entropy. Moreover, the results do not carry universal conclusions for general scientific audience or practitioners. The article cannot been accepted for publication in this form.
References
Strupczewski W.G., Kochanek K., Singh V.P. and Weglarczyk S. (2005) Are Parsimonious Flood Frequency Models More Reliable than the True Ones? I. Accuracy of Quantiles and Moments Estimation (AQME) – Method of Assessment. Acta Geophysica Polonica, Vol. 53, no 4, pp. 419-436
Kochanek K., Strupczewski W.G., Singh V.P. and Weglarczyk S. (2005) Are Parsimonious Flood Frequency Models More Reliable than the True Ones? II. Comparative assessment of the performance of simple models versus the parent distribution. Acta Geophysica Polonica, Vol. 53, no 4, pp. 437-457.
Author Response

(The authors gave the same response as above.)

Round 2
Reviewer 3 Report
Dear Authors,
Thank you very much for considering my remarks in a revised version of the manuscript. Now it reads good and can be published as it is.
With best regards